# FUT2 Facilitates Autophagy and Suppresses Apoptosis via p53 and JNK Signaling in Lung Adenocarcinoma Cells

**DOI:** 10.3390/cells11244031

**Published:** 2022-12-13

**Authors:** Yuqi Zhang, Enze Yao, Yijing Liu, Yining Zhang, Mengyang Ding, Jingyu Liu, Xiaoming Chen, Sairong Fan

**Affiliations:** 1Institute of Glycobiological Engineering, School of Laboratory Medicine and Life Sciences, Wenzhou Medical University, Wenzhou 325035, China; 2Key Laboratory of Laboratory Medicine, Ministry of Education, Wenzhou Key Laboratory of Cancer Pathogenesis and Translation, School of Laboratory Medicine and Life Sciences, Wenzhou Medical University, Wenzhou 325035, China; 3Department of Clinical Genetics, Northwest Women’s and Children’s Hospital, Xi’an 710061, China; 4Department of Laboratory, Zhoukou Central Hospital, Zhoukou 466000, China

**Keywords:** α-1,2-fucosyltransferase 2 (FUT2), autophagy, AMPK/ULK1, JNK, lung adenocarcinoma

## Abstract

Lung cancer is the most common cancer with high morbidity and mortality worldwide. Our previous studies showed that fucosyltransferase 2 (FUT2) is highly expressed in lung adenocarcinoma (LUAD) and plays a vital role in the tumorigenesis of LUAD. However, the underlying mechanism is not fully understood. Autophagy has recently attracted increasing attention due to its pro-survival role in cancer progression and metastasis. Here, we found that FUT2 was up-regulated and had an AUC (Area Under Curve) value of 0.964 in lung adenocarcinoma based on the TCGA dataset. Knockdown of FUT2 weakened the autophagy response, as evidenced by a degradation of LC3-II and Beclin1. The phosphorylation levels of AMPK, ULK1, and PI3K III were significantly reduced by FUT2 knockdown. FUT2 promoted the translocation of p53 from the cytoplasm into the nucleus, which triggered the DRAM1 pathway and enhanced autophagy. Meanwhile, the knockdown of FUT2 increased the phosphorylation of JNK and promoted mitochondrial-mediated apoptosis. Furthermore, the knockdown of FUT2 inhibited the autophagy induced by Z-VAD-FMK and promoted the apoptosis suppressed by rapamycin. The autophagy and apoptosis regulated by FUT2 antagonized each other. Taken together, these findings provide a mechanistic understanding of how FUT2 mediated the crosstalk between autophagy and apoptosis, which determine lung cancer cell death and survival, leading to the progression of lung adenocarcinoma.

## 1. Introduction

Lung cancer is the leading cause of cancer-related deaths and is among the most commonly diagnosed cancers worldwide, representing approximately one in five (18.0%) cancer deaths, according to global cancer statistics of 2020 [1,2]. Despite improvements in the detection techniques and remarkable therapeutic strategies in recent decades, the survival rates for lung cancer remain unfavorable. The 5-year survival of patients with stage I lung cancer is 56.3%, while the 5-year survival of patients with stage IV is only 4.7% [3]. Therefore, it is essential to explore the pathological mechanism of lung cancer progression and effective therapeutic targets.

Glycosylation, a basic and vital post-translational modification of proteins, plays important roles in almost all steps of tumor progression, and aberrant glycosylation is recognized as a hallmark of cancer [4,5]. As a type of glycosylation, fucosylation is catalyzed by fucosyltransferases (FUTs), which transfer the fucose residue to glycan chains of glycoproteins or glycolipids [6]. As a universal marker of tumor progression, deficient or excessive expression of FUTs greatly affects cell survival, adhesion, angiogenesis, immune evasion, and the metabolism of tumors, eventually resulting in the development of cancer [7]. Studies have demonstrated that several fucosyltransferases promote lung cancer progression. For instance, the overexpression of FUT8 is associated with unfavorable clinical outcomes in patients and mediates the malignant phenotypes of non-small cell lung cancer [8]. FUT4 is overexpressed in lung adenocarcinoma (LUAD) and is associated with the poor overall survival of LUAD patients, and a high expression of FUT4 promotes the metastasis of lung adenocarcinoma [9]. FUT2, together with FUT1, is responsible for the chemical synthesis of A, B, and H antigens in erythroid cell lines and secreted epithelial cell lines [10]. Recently, FUT2 has been shown to be involved in cancer progression, including cancer cell proliferation, invasion, metastasis, and cell signaling [10,11]. Our previous studies showed that FUT2 can promote lung adenocarcinoma cell proliferation, migration, and invasion and functions as a negative regulator of apoptosis [12,13]. However, the underlying mechanism of FUT2 in LUAD is not fully understood.

Autophagy, the evolutionarily conserved form of programmed cell death (PCD), plays a dual role in cancer: inhibiting tumor initiation or promoting tumor progression [14]. On the one hand, autophagy can function as a suppressor for tumor initiation through the removal of oncogenic proteins and damaged organelles, maintain genome stability, and work with immune surveillance to prevent inflammation [15]. On the other hand, once the tumors are fully established, autophagy can prevent cancer cell damage, promote cancer metastasis, and inhibit cancer therapy, thereby sustaining tumor cell survival and favoring tumor growth [16]. Growing evidence shows that autophagy plays a pro-survival role in cancer, and the inhibition of autophagy represents an attractive and potent strategy for cancer therapy [16]. Recent reports suggest that glycosylation regulates the process of autophagy. ULK1 (Unc-51-like kinase1) regulates the initiation of autophagy, and posttranslational modifications on ULK1 serve as a nexus for the control of autophagic induction [17,18]. Studies have indicated that O-GlcNAcylation of ULK1 is important for the activation of lipid kinaseVPS34, leading to the conversion of phosphatidylinositol-(3)-phosphate (PI(3)P) and phagophore formation [18]. Knock-down of ST8SIA1 reduced autophagosome levels by suppressing GD3 synthesis [19]. Sec24p and other CopII proteins are modified by O-GlcNAc and respond to pro-autophagic stimuli [20]. In addition, as powerful regulators of cell death or survival, autophagy and apoptosis can be interconnected and regulated by the same regulators, including Bcl-2, Beclin1, and p53 [21,22]. The interaction between autophagy and apoptosis determines the fates of cancer cells and may create opportunities for novel therapeutic strategies in cancer.

Here, we showed that FUT2 was overexpressed in lung adenocarcinoma and was associated with the T stage of lung adenocarcinoma. FUT2 can enhance ULK1 phosphorylation and p53 nuclear translocation and promotes autophagy by activating the AMPK/ULK1/PI3K III and p53/DRAM1 axes. Meanwhile, FUT2 can suppress apoptosis by inhibiting JNK signaling. The autophagy and apoptosis regulated by FUT2 antagonize each other. These results revealed the crosstalk between autophagy and apoptosis regulated by FUT2 in LUAD, providing new insight into the roles of FUT2 in tumorigenesis and tumor development, which might be helpful to identify diagnostic markers and therapeutic targets for lung adenocarcinoma.

## 2. Materials and Methods

### 2.1. Reagents, Plasmids, and Antibodies

SP600125 (HY-12041), wortmannin (HY-10197), rapamycin (HY-10219), chloroquine (CHQ) (HY-17589A), MG132 (HY-13259), and Z-VAD-FMK (HY-16658B) were purchased from MedChemExpres (Monmouth Junction, NJ, USA). Cycloheximide (CHX) (S7418) was purchased from Selleck (Shanghai, China). Psi-U6.1/eGFP/Puro-FUT2 and psi-U6.1/eGFP/Puro-NC vectors were designed by GeneCopoeia (Guangzhou, China). StubRFP-sensGFP-LC3 Lentivirus and FUT2 cDNA expression and control vectors were constructed by Shanghai Genechem. SiDRAM1 (C6133) was purchased from Shanghai GenePharma. Antibodies for LC3-II (18725-1-AP), Apaf-1 (21710-1-AP), Bcl-2 (12789-1-AP), cleaved Caspase-9 (10380-1-AP), p53 (10442-1-AP), and GAPDH (10494-1-AP) were purchased from Proteintech (Chicago, IL, USA). Antibodies specific against FUT2 (sc-100742), p-JNK^Tyr185, Thr183^ (81E11), JNK (sc-7345) were obtained from Santa Cruz Biotechnology (Santa Cruz, CA, USA). Antibodies for AMPK (db1167), Cytc (db2293), ULK1 (db2872), Tubulin (db3285), and β-actin (db10001) were purchased from Diagbio (Hangzhou, China). p-PI3K III (#13857) was purchased from Cell Signaling Technology (Danvers, MA, USA). Others antibodies for Beclin1 (ab207612), p62 (ab207305), p-AMPK^Thr183, Thr172^ (ab133448), p-ULK1 (ab133747), Atg5 (ab108327), DRAM1 (ab64739), and pro Caspase-9 (ab32539) were purchased from Abcam (Boston, MA, USA).

### 2.2. Bioinformatics Analysis

The TCGA_LUAD_L3 HTSeq-FPKM RNAseq data format of FUT2 and clinical information were downloaded from the TCGA database (https://portal.gdc.cancer.gov/ (accessed on 1 April 2021)) and converted to TPM format. TPM format RNA-seq data were obtained from the TCGA and Genotype-Tissue Expression (GTEx) database and were uniformly processed by Toil process from UCSC Xena (https://xenabrowser.net/datapages/ (accessed on 1 April 2021)) [23]. The log2 [TPM+1]-transformed expression data were applied to compare the FUT2 mRNA expression and clinical parameters in patients with lung adenocarcinoma, including pathological stages and TNM stage. Statistical analyses were performed using R package (V 3.6.3) gplot2. A paired *t*-test and the Mann–Whitney U-test were used to analyze the expression differences. The ROC curve was conducted using the pROC package. The relationship between FUT2 and immunocytes was detected by the GSVA package [24].

### 2.3. Cell Culture and Transfection

The lung cancer cell lines (A549, H1299, H460) and the normal pulmonary epithelial cell line (Beas-2b) were purchased from the American Type Culture Collection (ATCC). A549, H460, and Beas-2b cells were maintained in DMEM medium (high glucose, Sigma, St. Louis, MO, USA) with 10% FBS (10270-106, Gibco, Carlsbad, CA, USA). H1299 cells were maintained in RPMI-1640 medium (high glucose, Sigma, St. Louis, MO, USA) with 10% FBS. Cells were cultured in an incubator at 37 °C with 5% CO_2_. For transfection, the vectors (Psi-U6.1/eGFP/Puro-FUT2 and psi-U6.1/eGFP/Puro-NC) were transfected into A549 cells using Lipofectamine 3000 according to the manufacturer’s instructions. The cells were cultured in DMEM containing 10% FBS with 600 ng/mL puromycin under 5% CO_2_ at 37 °C for 2 weeks. To upregulate FUT2 expression, the FUT2 cDNA expression vector (Gemma, Shanghai, China) was transfected into A549 cells using Lipofectamine 3000 Transfection Reagent. The efficiency of FUT2 knockdown (or upregulation) was verified by reverse transcription-quantitative polymerase chain reaction (RT-qPCR) and Western blot analyses. The positive clones were selected for subsequent experiments.

### 2.4. RNA Extraction and Quantitative RT-PCR (RT-qPCR)

The total RNA was extracted with TRIzol reagent according to the manufacturer’s instructions. The cDNAs were obtained using a reverse transcriptase kit (RR047A, Takara, Beijing, China). Obtained cDNAs were quantified with an SYBR^®^ Premix Ex Taq^TM^ (Perfect Real Time) qPCR Kit (Takara, Beijing, China) and detected by an ABI real-time fluorescent quantitative PCR system (Applied Biosystems, Waltham, MA, USA). The 2^−ΔΔCt^ method was used to calculate the relative gene expression, and GAPDH was used as the internal reference. All experiments were performed in triplicate and were repeated three times. The RT-qPCR primers are listed in Table 1.

### 2.5. Western Blot Analysis

Cells were lysed in RIPA lysis buffer (Beyotime, Shanghai, China) supplemented with 1 mM PMSF (Beyotime, Shanghai, China). The concentration of protein was quantified by the BCA Protein Assay Kit (Beyotime, Shanghai, China). An equal amount of protein was separated by 10% sodium dodecyl sulfate polyacrylamide gel electrophoresis (SDS-PAGE). The proteins were then transferred onto a PVDF membrane (Millipore Corp, Burlington, MA, USA). After the membranes were blocked using Tris-buffered saline (TBST) with 5% nonfat milk and 0.1% Tween-20, the membranes were incubated with the first antibody (4 °C, overnight). After washing with TBST three times, the membranes were incubated with the corresponding second antibody (room temperature, 1 h) and detected using an enhanced chemiluminescence (ECL) reagent (NCM, Suzhou, Jiangsu, China). The results were analyzed by Image J. For Beclin1 protein degradation assay, cells (3 × 10^5^ cells/well) were spread in a 6-well plate and cultured overnight and then treated with CHX (100 μM) or CHX (100 μM) together with CHQ (25 μM) or MG132 (10 μM) for the indicated time periods. Treated cells were harvested and lysed. The proteins were extracted and analyzed by Western blot.

### 2.6. Autophagy Activity Detection

After the climbing tablets were placed in 24-well plates, the cells were incubated at a density of 15,000/well for virus infection (the virus concentration was 1 × 10^8^ TU/mL, the infection reagent was Hitrans G P, and the MOI value was 100). After 72 h, the culture medium was discarded, and cells were fixed with 4% paraformaldehyde (500 μL/holes) for 30 min at room temperature; then, the cells were stained with DAPI for 5 min and examined using a confocal laser scanning microscope.

### 2.7. Transcriptome Analysis

Transcriptome sequencing was based on the Illumina sequencing platform to analyze the gene expressions of A549 RNAi-FUT2 cells and NC cells. DEGs of autophagy and apoptosis were captured (*p* < 0.05).

### 2.8. Apoptosis Detection

Apoptosis was detected by an apoptosis kit (KAG107, KeyGEN BioTECH, Nanjing, Jiangsu, China) according to the manufacturer’s instruction. The cells were spread in a 6-well plate with a density of 2 × 10^5^ cells/well. wortmannin and rapamycin were added for the corresponding time, and cells were digested with 500 μL trypsin (C0205, Beyotime, Shanghai, China) without EDTA. The cells were blown by PBS to a form suspension, followed by centrifugation at 2000 rpm for 5 min, and the cell precipitates were obtained. The cells were stained with FITC-conjugated annexin V and Propidium Iodide (PI) and analyzed by a flow cytometer (Becton Dickinson, Franklin Lake, NJ, USA).

### 2.9. Mitochondrial Membrane Potential (MMP) Detection

The MMP was assessed according to the instructions of a mitochondrial membrane potential detection kit (C2006, Beyotime, Shanghai, China). The cells (2 × 10^5^ cells/well) were spread in a 6-well plate and cultured overnight. After incubation for 20 min at 37 °C, 1 × JC-1 staining buffer was added, and the cells were digested by trypsin. The cell precipitation was obtained by centrifugation and resuscitated with 300 μL staining buffer and detected by flow cytometry.

### 2.10. Immunofluorescence

The processed cells were fixed in 4% paraformaldehyde at room temperature for 30 min. After being washed with PBS, cells were permeabilized with 0.5% Tritonx-100 for 15 min. Then, the corresponding primary antibodies were added, and the samples were incubated at 4 °C overnight, followed by incubation with FITC-conjugated secondary antibodies for 1 h at room temperature without light. Finally, cells were stained with DAPI for 10 min and analyzed using a confocal laser scanning microscope.

### 2.11. Co-Immunoprecipitation (Co-IP)

Co-IP was used to investigate the protein interactions. Briefly, the pretreated cells were lysed as mentioned above. The supernatants were collected and pre-cleared with Protein A+G Agarose beads (P2055, Beyotime, Shanghai, China) and incubated with the corresponding primary antibody or mouse IgG (A7028, Beyotime, Shanghai, China) at 4 °C overnight. Then, the sample was incubated with pre-washed Protein A+G Agarose beads at 4 °C for 2 h. The protein–antibody complexes were collected by centrifugation at 3000 rpm for 10 min, followed by washing with PBS. Immunoprecipitated proteins were eluted from the beads in SDS sample loading buffer for 10 min at 100 °C. The samples were analyzed by Western blot.

### 2.12. In Vivo Experiment

BALB/c-nu nude mice (6-week-old, female) were purchased from the Beijing Vital River Laboratory Animal Technology Co., Ltd. (Beijing, China). Mice were randomly assigned to experimental groups and housed at ~25 °C with a 12 h light and dark cycle and maintained on food and water ad libitum. For the in vivo subcutaneous tumorigenesis assay, A549, NC, and RNAi-FUT2 A549 cells (1 × 10^7^ cells in 100 μL DMEM) were injected into the subcutaneous right flanks of the BALB/C-nu nude mice. The mice were euthanized after 30 days, and the implanted tumors were removed. The animal experiment was approved by the Laboratory Animal Ethics Committee of Wenzhou Medical University and Laboratory Animal Centre of Wenzhou Medical University.

### 2.13. Statistical Analysis

Statistical analyses were performed using SPSS software version 17.0 (Chicago, IL, USA). Student’s *t*-test was used to compare between two groups in the in vitro experiments. Dunnett’s multiple comparison test was used if two groups did not have equal variance. Data were expressed as the mean value ± S.D. of at least three repeated experiments, and *p*-values less than 0.05 were considered to be statistically significant.

## 3. Results

### 3.1. Upregulated mRNA Expression of FUT2 in Patients with Lung Adenocarcinoma

To clarify the role of FUT2 in lung cancer, FUT2 expression levels in different cancer types were analyzed using the TCGA database. As shown in Figure 1A, FUT2 was significantly upregulated in seven cancer types but downregulated in six cancer types compared with normal tissues. The results showed that the levels of FUT2 were significantly higher in lung adenocarcinoma (Figure 1B, 4.111 ± 1.143 vs. 1.619 ± 0.667, Weltch *t*-test, *p* < 0.001. Figure 1C, 3.539 ± 1.126 vs. 0.942 ± 0.809, Weltch *t*-test, *p* < 0.001) than in the normal control. Paired data analysis indicated the same results (Figure 1D, 3.978 ± 1.134 vs. 1.626 ± 0.674, *p* < 0.001, Normal = 57 and Tumor = 57).

To clarify the value of FUT2 in diagnosing lung adenocarcinoma, a ROC curve analysis was conducted. The results showed that FUT2 had an AUC value of 0.964 (95% CI: 0.949–0.979) (Figure 1E). At a cutoff of 2.857, FUT2 had a sensitivity, specificity, and accuracy of 96.6, 88.2, and 89.1%, respectively. The positive predictive value was 47.5%, and the negative predictive value was 99.6%. These findings indicated that FUT2 could be a potential biomarker for lung adenocarcinoma.

### 3.2. Relationships between the mRNA Levels of FUT2 and Clinical Pathological Characteristics of Lung Adenocarcinoma Patients

To investigate the relationships between the FUT2 expression and clinical pathological characteristics of lung adenocarcinoma, the data of 535 tumors and 59 normal samples in TCGA were downloaded and analyzed using a Chi-square test and the Wilcoxon rank sum test. As shown in Table 2, the expression of FUT2 was higher in the T stage (*p* = 0.001), but there was no correlation with the pathological stage, N stage, M stage, and residual tumor.

### 3.3. FUT2 and LC3-II Were Highly Expressed in Lung Adenocarcinoma Cell Lines

To validate the expression of FUT2 in lung cancer cell lines, A549, H1299, H460, and Beas-2B cells were used. The results showed that the expression of FUT2 was significantly higher in lung cancer cells (A549, H1299, and H460) than in Beas-2B cells, especially in A549 cells (Figure 2A). LC3-II, which is associated with the extent of autophagosome formation, was measured. The results showed that the expression of LC3-II was higher in lung cancer cells (A549, H1299, and H460) than in Beas-2B cells, which is consistent with the expression of FUT2 in these cell lines. This finding suggested that the expression of FUT2 might be associated with autophagy in lung adenocarcinoma.

### 3.4. FUT2 Promoted the Autophagy of Lung Adenocarcinoma Cells

To evaluate the effect of FUT2 on the autophagy of lung adenocarcinoma cells, a specific shRNA against FUT2 was transferred into LUAD cells (A549 and H1299) to silence FUT2 expression in LUAD cells, which was assessed by RT-qPCR and Western blot. The mRNA and protein levels of FUT2 were dramatically reduced in RNAi-FUT2 cells compared with A549 or NC cells (scrambled control) (Figure 2B,C). As shown in Figure 2D, the expressions of Beclin1, Atg5, and LC3-II, which are autophagy-related proteins, were significantly reduced in FUT2-knockdown A549 cells compared with those in NC control cells. The expressions of Atg5 and LC3-II were significantly reduced in FUT2 knockdown H1299 cells compared with those in NC control cells (Figure 2D). Meanwhile, the mRNA levels of LC3-II and Beclin1 were also significantly decreased in the RNAi-FUT2 A549 group compared with the NC group (Figure 2E). Interestingly, the protein and mRNA levels of p62, a marker of the autophagy explanatory phase, were also markedly reduced by FUT2 knockdown, suggesting that the knockdown of FUT2 blocks autophagic flux (Figure 2D,E). The expressions of Atg5 and LC3-II increased in FUT2 overexpression A549 cells (Figure 2F). The expressions of LC3-II, p62, and Atg5 increased significantly in FUT2 overexpression H460 cells compared with those in H460 and vector control cells (Figure 2G). We carried out further verification of the effect of FUT2 on autophagy in subcutaneous tumor models. The results showed that the expressions of FUT2, LC3-II, and p62 decreased in the tumor tissues of the FUT2 knockdown group compared with the A549 and NC groups (Figure 2H). This was consistent with the in vitro results. These results indicated that FUT2 promotes the autophagy of LUAD cells.

### 3.5. FUT2 Affected the Early Stage of Autophagy in Lung Adenocarcinoma Cells

When cells lack nutrients, autophagy is enhanced to maintain cellular homeostasis [25]. As shown in Figure 3A, starvation, commonly used to stimulate autophagy, significantly increased the LC3-II levels in the NC and RNAi-FUT2 cells compared to the corresponding cells maintained in normal medium (Figure 3A). The knockdown of FUT2 markedly reduced the expression of LC3-II with or without starvation treatment compared to the corresponding control cells (Figure 3A). Ammonium chloride (NH_4_Cl), the early-stage inhibitor of autophagy, was used to treat the cells. The expression of LC3-II was increased in the NH_4_Cl-treated groups compared with the corresponding control group (without NH_4_Cl-treated) (Figure 3B). Knockdown of FUT2 suppressed the expression of LC3-II induced by NH_4_Cl. The results suggested that the autophagy induced by starvation or NH_4_Cl can be suppressed by FUT2 knockdown, indicating that FUT2 may be involved in the early stage of autophagy in A549 cells.

To further confirm the effect of FUT2 on autophagy, we utilized the stubRFP-sensGFP-LC3 double-fluorescence system, which is a prominent way to visually detect the activity of autophagy. The fluorescence effect was greater with the use of the infection reagent at the MOI value of 100 (Figure 3C), and we selected this condition to further evaluate the changes in autophagy flux. As expected, the knockdown of FUT2 significantly decreased both LC3-II yellow puncta (representing autophagosomes) and red-only puncta (representing autolysosomes) in A549 cells (Figure 3D). The findings suggested that the knockdown of FUT2 suppressed autophagy in A549 cells.

### 3.6. FUT2 Promotes Autophagy by AMPK/ULK1/PI3K III Pathways in LUAD Cells

Autophagy is initiated by ULK1, which can be phosphorylated by AMPK [26]. Based on this, we investigated the effect of FUT2 on the AMPK/ULK1 signaling pathways. As shown in Figure 4A, the knockdown of FUT2 did not affect the expressions of ULK1 and AMPK, whereas the levels of p-ULK1 and p-AMPK significantly decreased in the FUT2 knockdown cells, indicating that FUT2 promoted the phosphorylation of ULK1 and AMPK. Meanwhile, the level of p-PI3K III, which combines with Beclin1 to form class III PI3K-Beclin1 complexes, was also reduced by FUT2 knockdown.

As mentioned above, the knockdown of FUT2 decreased the protein level of Beclin1. To further ascertain how FUT2 stabilizes the Beclin1 protein, protein synthesis inhibitor Cycloheximide (CHX), lysosome inhibitor Chloroquine (CHQ), and proteasome inhibitor MG-132 were used to test whether Beclin1 degradation occurred through the ubiquitin–proteasome system or autophagy–lysosome pathway or both. The results showed that Beclin1 abundance was decreased by CHX treatment, and this reduction was further decreased by FUT2 knockdown (Figure 4B). MG132 but not CHQ treatment prevented the degradation of Beclin1 upon knockdown of FUT2 (Figure 4B), suggesting that FUT2 stabilizes Beclin1 by protecting Beclin1 from proteasome degradation.

### 3.7. FUT2 Stimulates Autophagy through p53/DRAM1 Signaling in LUAD Cells

To further investigate the underlying mechanism of FUT2 participating in autophagy, the transcriptome of A549 cells was examined. As shown in Figure 5A, the mRNA levels of ULK1 and Atg101 were reduced in FUT2 knockdown cells compared with those in NC cells. The mRNA level of NBR1, which is involved in autophagy degradation, was also decreased by FUT2 knockdown. In addition, the expression of DRAM1 (DNA damage regulated autophagy modulator 1) was decreased, and TAB3 (an autophagy negatively regulated gene) was increased by FUT2 knockdown.

The Western blot results confirmed that DRAM1 was significantly decreased by FUT2 knockdown (Figure 5B), and Co-IP showed that FUT2 interacted with DRAM1 (Figure 5F). DRAM1 is a downstream target molecule of p53 and induces autophagy by nuclear p53 [27]. The effect of FUT2 on p53 distribution was analyzed by cell fractionation assay. The results showed that the nucleus fraction of p53 was drastically reduced and the cytoplasmic p53 was significantly elevated in FUT2 knockdown cells compared with the control cells (A549 and NC) (Figure 5C), suggesting that the knockdown of FUT2 suppressed p53 translocation into the nucleus in A549 cells. The distribution of p53 regulated by FUT2 was further confirmed by immunofluorescence in A549 cells (Figure 5D). In order to examine whether DRAM1 is responsible for FUT2-induced autophagy in A549 cells, siRNA against DRAM1 was used to silence the expression of DRAM1. The results showed that the expressions of DRAM1 and LC3-II were significantly reduced by siDRAM1 or FUT2 knockdown compared with their corresponding control group. Simultaneous down-regulation of FUT2 and DRAM1 significantly increased the reducing effect of FUT2 or DRAM1 knockdown on the expression of LC3-II in A549 cells (Figure 5E). Furthermore, the Co-IP results showed that FUT2 interacted with p53 (Figure 5F), which might prevent the nuclear translocation of p53. These findings suggested that p53/DRAM1 signaling was involved in the autophagy induced by FUT2.

### 3.8. FUT2 Inhibited the Mitochondrial-Mediated Apoptosis in LUAD Cells

As programmed cell death, autophagy and apoptosis have now been shown to be interconnected and implicated in tumor cells’ survival or death. Our previous study indicated that FUT2 suppresses the apoptosis of lung adenocarcinoma cells [12]; however, the underlying mechanism is still unclear. Based on the transcriptome results, the mRNA levels of CCAR1 (cell division cycle and apoptosis regulator 1) and BCLAF1 (BCL2-associated transcription factor 1) increased in the FUT2 knockdown cells, and the mRNA levels of UQCR10 (ubiquinol–cytochrome c reductase, a subunit of mitochondrial complex III) and RAPR9 decreased in the FUT2 knockdown cells compared with those in the control cells (Figure 5A), suggesting that FUT2 is involved in the apoptosis of A549 cells.

To assess the role of FUT2 in apoptosis, we examined the expressions of intrinsic apoptosis relative proteins, such as Apaf-1, Caspase-9, and cytochrome c (Cytc) by Western blot. The results showed that the expressions of Apaf-1, Cytc, pro Caspase 9, and cleaved Caspase 9 were significantly increased, and the expression of Bcl-2 was decreased in FUT2-knockdown A549 cells compared with A549 or NC cells (Figure 6A). The expressions of Apaf-1, cleaved Caspase 9, and Cytc were significantly increased in FUT2 knockdown H1299 cells compared with the control cells (Figure 6A). Meanwhile, the expressions of Apaf-1, cleaved Caspase 9, and Cytc were significantly reduced in FUT2-upregulated H460 cells compared with the control cells (Figure 6B). Moreover, the localization of Cytc was examined by immunofluorescence assays. As showed in Figure 6C, there was a significant reduction in yellow fluorescence in the FUT2-knockdown A549 cells compared to the NC cells, indicating that FUT2 reduced the accumulation of Cytc in the mitochondria. It is generally accepted that the decrease in the mitochondrial membrane potential (MMP) is a symbolic event in early apoptosis. The effect of FUT2 to depolarize the mitochondrial membrane was investigated by JC-1 staining, and then fluorescence was detected using flow cytometry. As shown in Figure 6D, MMP was significantly reduced in FUT2 knockdown cells compared with the NC cells, suggesting that the knockdown of FUT2 promoted the damage caused to the mitochondria. These results suggested that FUT2 regulated the apoptosis of LUAD cells via the mitochondrial pathway.

### 3.9. FUT2 Modulates Apoptosis via JNK Signaling in LUAD Cells

To investigate the mechanism of FUT2 in apoptosis, the JNK signaling pathway, which is closely related to tumor cell apoptosis, was examined. As shown in Figure 6E, the knockdown of FUT2 did not affect the expression of JNK; however, the level of p-JNK (TRY185) was significantly increased in FUT2-knockdown A549 cells compared with NC cells. Next, SP600125, a specific inhibitor of the JNK signaling pathway, was used to characterize the role of FUT2 in the JNK pathway. After treatment with SP600125, Apaf-1 and p-JNK were significantly decreased compared with the corresponding cells without SP600125 treatment. Notably, the upregulation of Apaf-1 and p-JNK induced by FUT2 knockdown was markedly reduced by SP600125 treatment (Figure 6F). In addition, the Co-IP results showed that FUT2 interacted with JNK (Figure 6G), which might inhibit the phosphorylation of JNK. These results suggested that FUT2 can interact with JNK and inhibit the phosphorylation of JNK, leading to the suppression of apoptosis in A549 cells.

### 3.10. The Regulation of Autophagy and Apoptosis by FUT2 and Their Antagonism in LUAD Cells

Results showed that the expression of LC3-II was markedly elevated in SP600125-treated cells with or without FUT2 knockdown compared with their corresponding control cells (Figure 6F), indicating that FUT2 may involve the intricate interlinkage between autophagy and apoptosis in A549 cells. The knockdown of FUT2 enhanced the inhibition of the expression of LC3-II in wortmannin-treated cells (Figure 7A). Meanwhile, Apaf-1 was significantly increased in wortmannin-treated cells compared with the corresponding control group (Figure 7A). The apoptosis rates were significantly increased in wortmannin-treated cells compared with their corresponding control cells. The knockdown of FUT2 further increased the apoptosis rate induced by wortmannin (Figure 7B). The apoptosis rate was significantly decreased after the cells were treated with rapamycin, an autophagy activator, compared with the untreated cells (Figure 7B). Meanwhile, the apoptosis rate was markedly increased when DRAM1 expression was suppressed (Figure 7C). In addition, when cells were treated with Z-VAD-FMK, the expressions of Apaf-1 and cleaved caspase 9 decreased, and the expression of LC3-II increased compared with the corresponding control cells (Figure 7D). These results suggested that the autophagy and apoptosis regulated by FUT2 antagonized each other.

### 3.11. The Correlation between FUT2 Expression and Immune Infiltration

The Spearman correlation between the expression of FUT2 and the 24 immune cell types was analyzed by ssGSEA [28]. The results showed that higher FUT2 expression was positively associated with Th17 cells and negatively associated with T cells, cytotoxic cells, DC, Th1 cells, and T helper cells (*p* < 0.001, Figure 8A–H).

## 4. Discussion

FUT2, a Golgi stack membrane protein, participates in the synthesis of the H antigen and soluble A and B antigens [10]. In recent years, the roles of FUT2 in tumors have received more and more attention. Our previous studies showed that FUT2 is upregulated in lung adenocarcinoma tissues, is associated with poor survival, and plays an important role in the development of lung adenocarcinoma [12,13]. Here, we found that the expression of FUT2 was associated with T classification. ROC analysis revealed that FUT2 had a diagnostic value, providing evidence that FUT2 might be a potential biomarker for lung adenocarcinoma. However, the underlying mechanism of FUT2 in LUAD is still unclear.

As a type of programmed cell death, autophagy has been widely research in cancer. However, the dual role of autophagy in cancer progression and inhibition remains controversial [29]. Here, we found that the expression of LC3-II in lung cancer cell lines was correlated with the expression of FUT2. Autophagy is a strictly regulated system that is controlled by multiple key autophagy-related molecules including ULK1, Beclin1, Atg5, LC3, and so on [30]. The knockdown of FUT2 in LUAD cells reduced the expressions of LC3-II, Beclin1, Atg5, and p62 in vivo and in vitro and reduced the autophagy induced by starvation or NH_4_Cl. Meanwhile, the expressions of LC3-II and Atg5 were significantly increased in FUT2-upregulated A549 or H460 cells. These results suggest that FUT2 regulated the autophagy of LUAD cells. The mRFP-GFP-LC3 double labeling experiment further confirmed that the knockdown of FUT2 reduced the autophagy of A549 cells. These data suggested that the autophagy regulated by FUT2 might play a role in LUAD initiation and progression.

As key regulators of autophagy initiation and progression, both ULK1 and VPS34 are regulated by AMPK [31]. ULK1 plays a critical role in the initiation of autophagy and is activated by AMPK-mediated phosphorylation [26]. ULK1 also directly phosphorylates Beclin1 and recruits AMPK to VPS34 and Atg14, which allows AMPK to indirectly cooperate with Class III phosphoinositide 3-kinase (PI3K) to further condition autophagy [31,32]. Class III PI3K (also known as VPS34 in mammals) can associate with Beclin1, which binds to several proteins, leading to the promotion or inhibition of the autophagic function of VPS34 [31]. In the present study, the protein levels of AMPK and ULK1 were unaltered by FUT2 knockdown, while the phosphorylation of AMPK, ULK1, and PI3K III was markedly reduced. The knockdown of FUT2 promoted Beclin1 degradation through the proteasome pathway. These findings suggest that FUT2 regulates autophagy via the AMPK/ULK1 signaling pathway.

Apoptosis, as a classic mode of cell death, can suppress the growth and metastasis of cancer cells [33]. Intrinsic apoptosis, one of the main forms of apoptosis, can be initiated by the cell itself via a number of intracellular sensors released from the mitochondria [34]. Here, our results showed that the knockdown of FUT2 significantly increased the expressions of Apaf-1, pro-caspase-9, cleaved caspase-9, and Cytc in A549 and H1299 cells and decreased the expression of Bcl-2 in A549 cells. Meanwhile, the upregulated FUT2 significantly decreased the expressions of Apaf-1, cleaved caspase-9, and Cytc in H460 cells. Bcl-2, a classical anti-apoptotic protein, can inhibit intrinsic apoptosis by affecting the release of Cytc [35]. Additionally, the knockdown of FUT2 increased the accumulation of damaged mitochondria and the release of Cytc from the mitochondria. Cytc, released from mitochondria, binds to Apaf-1 to form apoptotic bodies, which recruit pro-caspase-9 to cause a caspase cascade reaction, and eventually apoptosis occurs [36]. Many studies point out that the JNK pathway has complex roles in mitochondria-mediated apoptosis [37]. As a response to endogenous apoptotic stimuli, JNK regulates the levels of Bcl-2 family proteins or their activities, which leads to a decrease in MMP and the release of Cytc, and these are important events in intrinsic apoptosis [38,39]. Our results showed that the knockdown of FUT2 increased the phosphorylation of JNK, decreased Bcl-2 and MMP, and released Cytc, which ultimately led to apoptosis. These findings suggested that FUT2 may inhibit the mitochondrial pathway of apoptosis to sustain the survival of A549 cells.

There are multiple connections between the autophagic and apoptotic processes, which seal the fate of the cell and play pivotal roles in tumor cell death and survival [22]. Here, we found that the apoptosis induced by FUT2 knockdown was suppressed by autophagy activation and enhanced by autophagy inhibition. Meanwhile, the autophagy reduced by FUT2 knockdown was increased by inhibiting apoptosis. These data suggest interplay between autophagy and apoptosis regulated by FUT2 in LUAD cells. FUT2 initially induces autophagy by exerting a cytoprotective function to prevent apoptosis in lung adenocarcinoma cells. Inversely, the regulation of apoptosis by FUT2 can antagonize the level of autophagy. Furthermore, we found that the knockdown of FUT2 significantly decreased the expression of DRAM1, which participates in the regulation of both cell autophagy and apoptosis [40]. The simultaneous knockdown of FUT2 and DRAM1 increased the decreasing effect of FUT2 knockdown on the expression of LC3-II. As a downstream molecule of p53, DRAM1 can promote autophagy by nuclear p53 [27]. The cytosolic pool of p53 represses autophagy by inhibiting the AMPK pathway, and the nuclear translocation of p53 facilitates the induction of autophagy [27,41]. Here, we found that FUT2 interacted with p53 and DRAM1, and the distribution of p53 was regulated by FUT2. The knockdown of FUT2 significantly decreased the nuclear translocation of p53 and increased cytosolic p53. RNA-seq analysis showed that the knockdown of FUT2 increased the expression of BCLAF1, which can promote p53-dependent apoptosis or promote apoptosis by inhibiting Bcl-2 expression [42]. It has been demonstrated that p53 can be triggered by activated JNK [37]. JNK can also regulate both autophagy and apoptosis by the phosphorylation and localization of Bcl-2 and Beclin1 [22]. In the present study, after treatment with SP600125, the expression of Apaf-1 decreased, and the expression of LC3-II increased with decreasing JNK phosphorylation levels. This suggests that FUT2 regulated apoptosis and autophagy through the JNK pathway in LUAD. FUT2 may induce autophagy through the cytoplasmic and nuclei distribution of p53, which regulates the AMPK-ULK1-PI3K III axis and DRAM1 signaling. Additionally, FUT2 suppresses apoptosis by inactivating JNK, which further promotes autophagy.

## 5. Conclusions

The present study further confirms that FUT2 serves as a positive regulator in lung adenocarcinoma development and highlights the role of FUT2 in the tumorigenesis of LUAD. FUT2 promotes lung adenocarcinoma progression by enhancing autophagy through the nuclear p53-DRAM1 and the plasma p53-AMPK-ULK1-PI3K III axes and decreases intrinsic apoptosis via the JNK-Cytc-Capase-9 axis. The study provides an integrated understanding of how FUT2 links autophagy and apoptosis in LUAD, which will be helpful for future diagnosis and therapy in lung adenocarcinoma.

## Figures and Tables

**Figure 1 cells-11-04031-f001:**
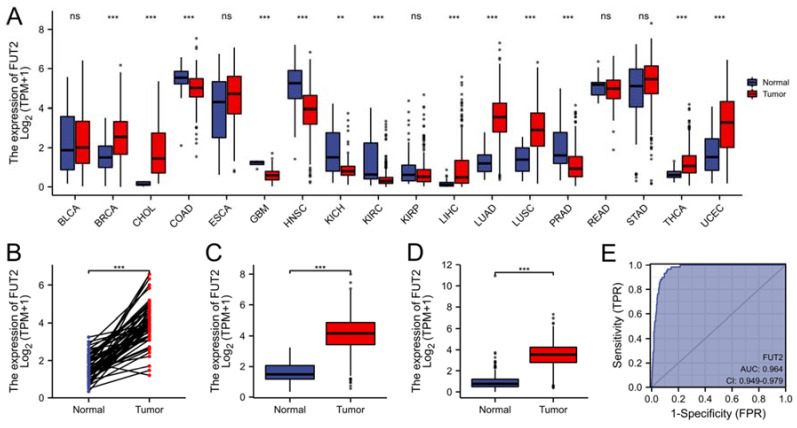
Bioinformatics analysis of FUT2 based on the TCGA and GTEx datasets. (**A**) The mRNA expression of FUT2 in different cancer types. (**B**,**C**) FUT2 expression in lung adenocarcinoma tissues and normal tissues in the TCGA dataset (**B**) and the TCGA + GTEx dataset (**C**). (**D**) FUT2 expression in lung adenocarcinoma and adjacent normal tissues. (**E**) ROC curve of FUT2 mRNA expression in the lung adenocarcinoma cohort (AUC = 0.964) (ns: No significance, ** *p* < 0.01, and *** *p* < 0.001).

**Figure 2 cells-11-04031-f002:**
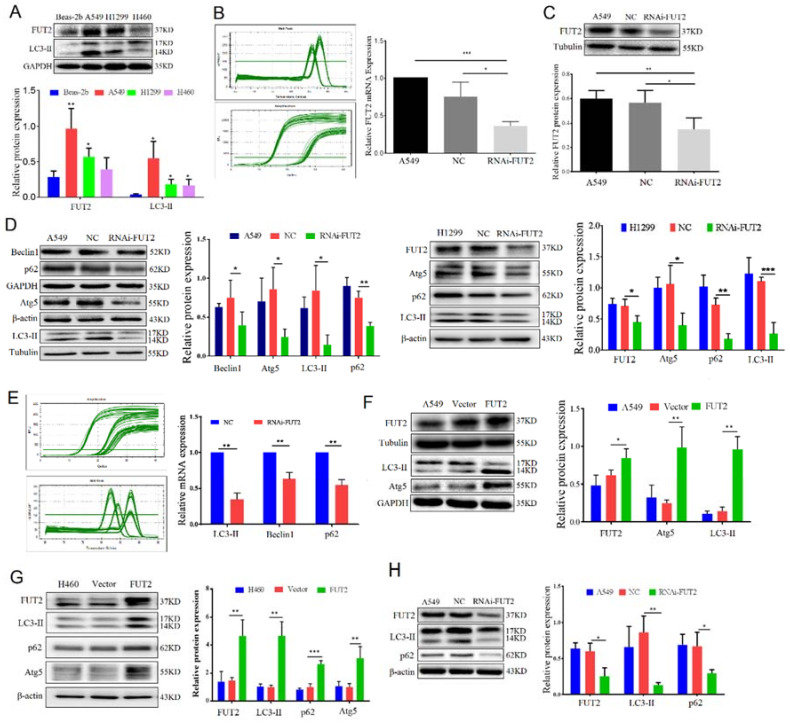
FUT2 promotes the autophagy of lung adenocarcinoma cells. (**A**) Representative Western blot showing the expressions of FUT2 and LC3-II in diverse lung cancer cell lines (A549, H1299, H460) and Beas-2b cells. (**B**,**C**) Representative RT-qPCR and Western blot showing successful constructions of RNAi-FUT2 A549 cells. (**D**) Representative Western blot showing that knockdown of FUT2 inhibited the expressions of Beclin1, Atg5, p62, and LC3-II in A549 and H1299 cells. (**E**) Representative RT-qPCR showing that the knockdown of FUT2 decreased the mRNA expression of LC3, Beclin1, and p62 in A549 cells. (**F**) Representative Western blot showing that the upregulation of FUT2 expression promoted the expressions of Atg5 and LC3-II in A549 cells. (**G**) Representative Western blot showing that the upregulation of FUT2 expression promoted the expressions of LC3-II, p62, and Atg5 in H460 cells. (**H**) Representative Western blot showing that the knockdown of FUT2 inhibited the expressions of LC3-II and p62 in tumors (* *p* < 0.05, ** *p* < 0.01, and *** *p* < 0.001).

**Figure 3 cells-11-04031-f003:**
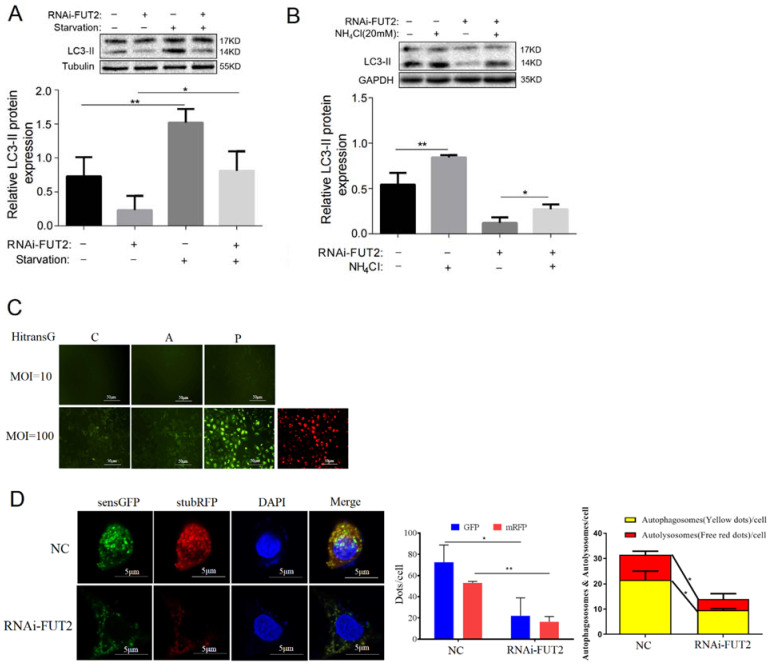
FUT2 boosts the early stage of autophagy in lung adenocarcinoma cells. (**A**) Representative Western blot showing that knockdown of FUT2 reduced the expression of LC3-II induced by serum starvation in A549 cells. (**B**) Representative Western blot showing that the knockdown of FUT2 reduced the expression of LC3-II induced by NH_4_Cl in A549 cells. (**C**) Representative image of immunofluorescence staining showing the stubRFP-sensGFP-LC3 double-fluorescence system. (**D**) Representative image of immunofluorescence staining showing that the knockdown of FUT2 decreased both autophagosomes (yellow dots) and autolysosomes (free red dots). (* *p* < 0.05, and ** *p* < 0.01).

**Figure 4 cells-11-04031-f004:**
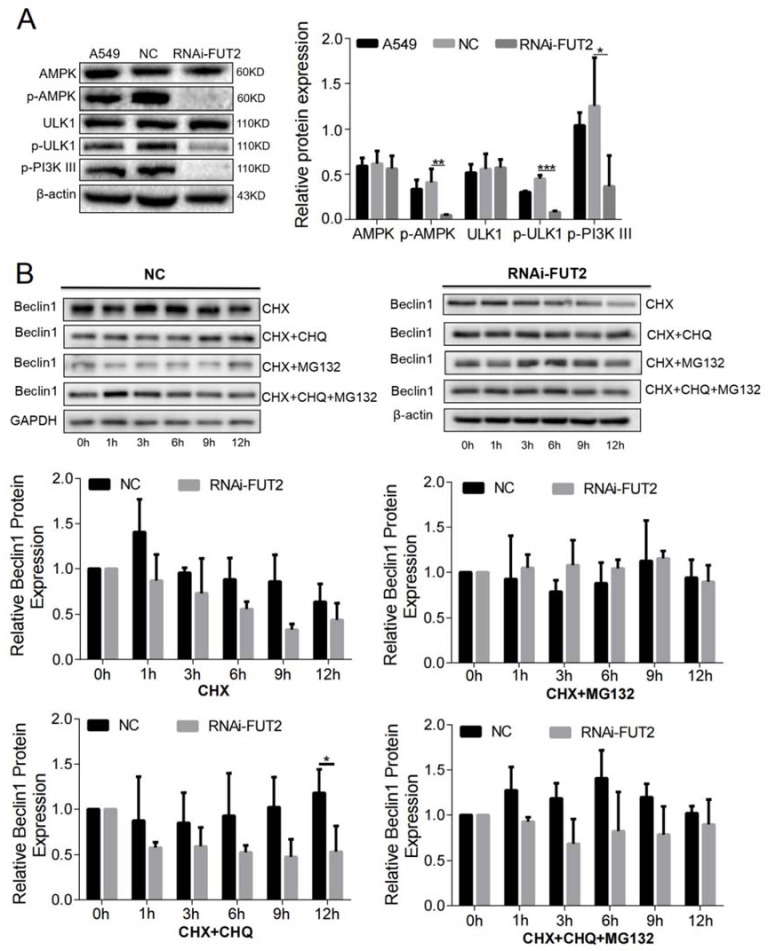
FUT2 promotes autophagy through the AMPK/ULK1/PI3K III pathway in A549 cells. (**A**) Representative Western blot showing that the knockdown of FUT2 reduced the levels of p-AMPK, p-ULK1, and p-PI3k III in A549 cells. (**B**) Representative Western blot showing the time course of the Beclin1 protein levels in FUT2-knockdown A549 cells or control cells according to different times in CHX, in the presence or absence of CHQ or MG132. (* *p* < 0.05, ** *p* < 0.01, and *** *p* < 0.001).

**Figure 5 cells-11-04031-f005:**
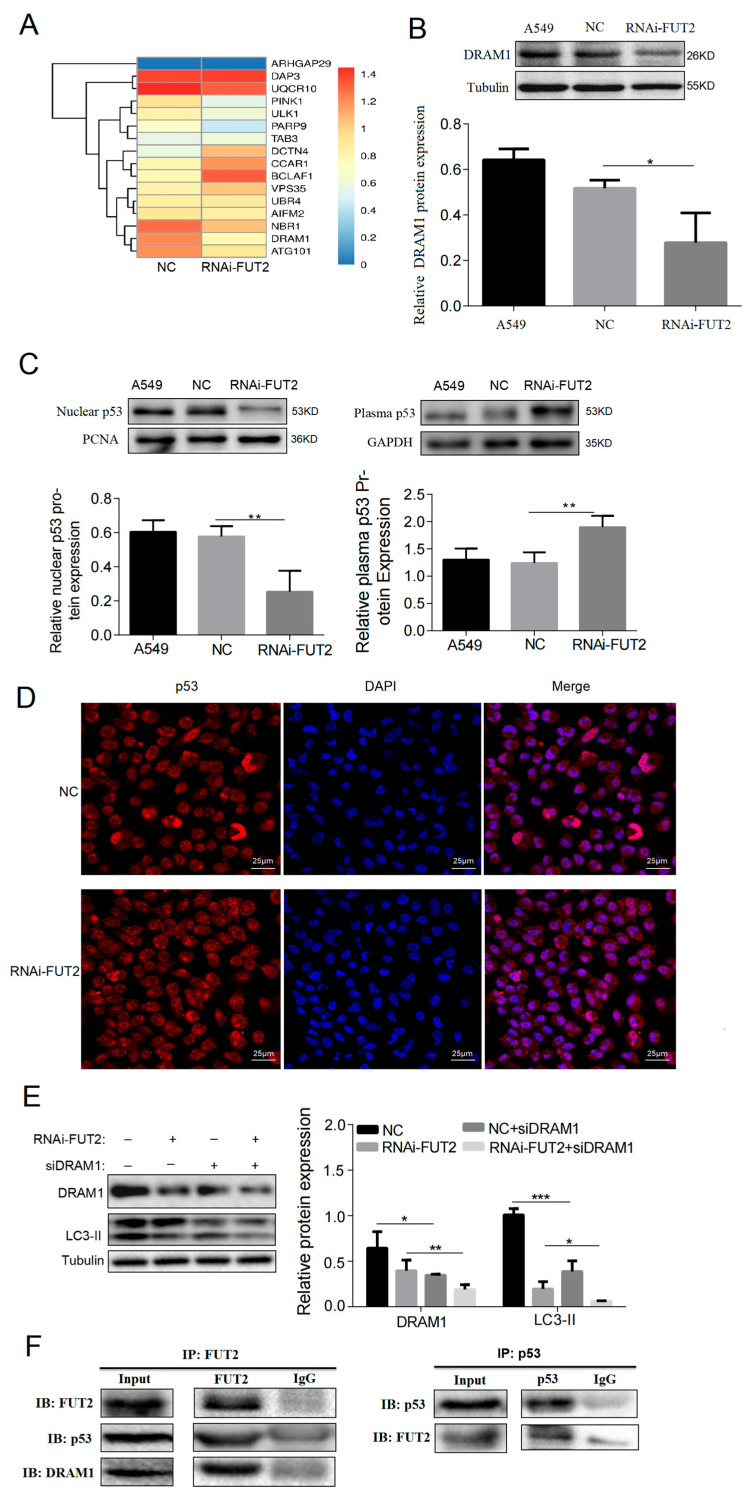
FUT2 stimulates autophagy through the nuclear p53-DRAM1 axis in A549 cells. (**A**) RNA-seq sequencing showing the genes associated with autophagy or apoptosis regulated by FUT2 in A549 cells. (**B**) Representative Western blot showing that the knockdown of FUT2 inhibited the expression of DRAM1 in A549 cells. (**C**) Representative Western blot showing the plasma and nuclear p53 protein levels in A549 cells. (**D**) Representative image of immunofluorescence staining showing the distribution of p53 in FUT2-knockdown A549 cells or control cells. (**E**) Representative Western blot showing the expressions of DRAM1 and LC3-II in FUT2-knockdown A549 cells or control cells in the presence or absence of siDRAM1. (**F**) Co-immunoprecipitation (Co-IP) assay showing an interaction between FUT2 and p53 or DRAM1 in A549 cells. (* *p* < 0.05, ** *p* < 0.01, and *** *p* < 0.001).

**Figure 6 cells-11-04031-f006:**
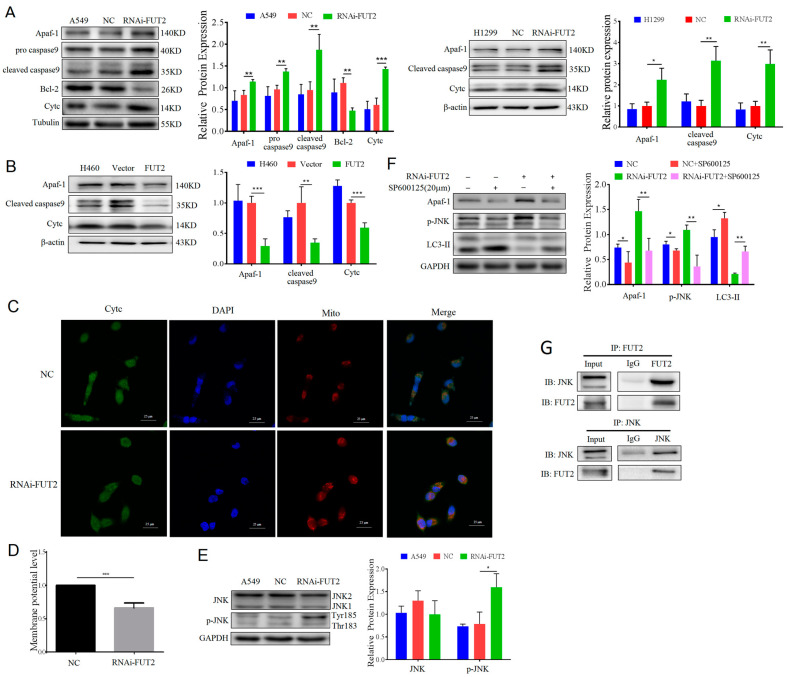
FUT2 depresses mitochondrial-mediated apoptosis in the JNK pathway in LUAD cells. (**A**) Representative Western blot showing that the knockdown of FUT2 increased the expressions of Apaf-1, pro caspase 9, cleaved caspase 9, Bcl-2, and Cytc in A549 and H1299 cells. (**B**) Representative Western blot showing that the upregulation of FUT2 expression reduced the expressions of Apaf-1, cleaved caspase 9, and Cytc in H460 cells. (**C**) Representative image of immunofluorescence staining showing the localization of Cytc in FUT2-knockdown A549 cells or control cells. (**D**) The MMP significantly decreased in RNAi-FUT2 A549 cells. (**E**) Representative Western blot showing that the knockdown of FUT2 increased the level of p-JNK. (**F**) Representative Western blot showing that the levels of Apaf-1, p-JNK, and LC3-II in FUT2-knockdown A549 cells or control cells in the presence and absence of SP600125. (**G**) Co-immunoprecipitation (Co-IP) assay showed an interaction between FUT2 and JNK in A549 cells. (* *p* < 0.05, ** *p* < 0.01, and *** *p* < 0.001).

**Figure 7 cells-11-04031-f007:**
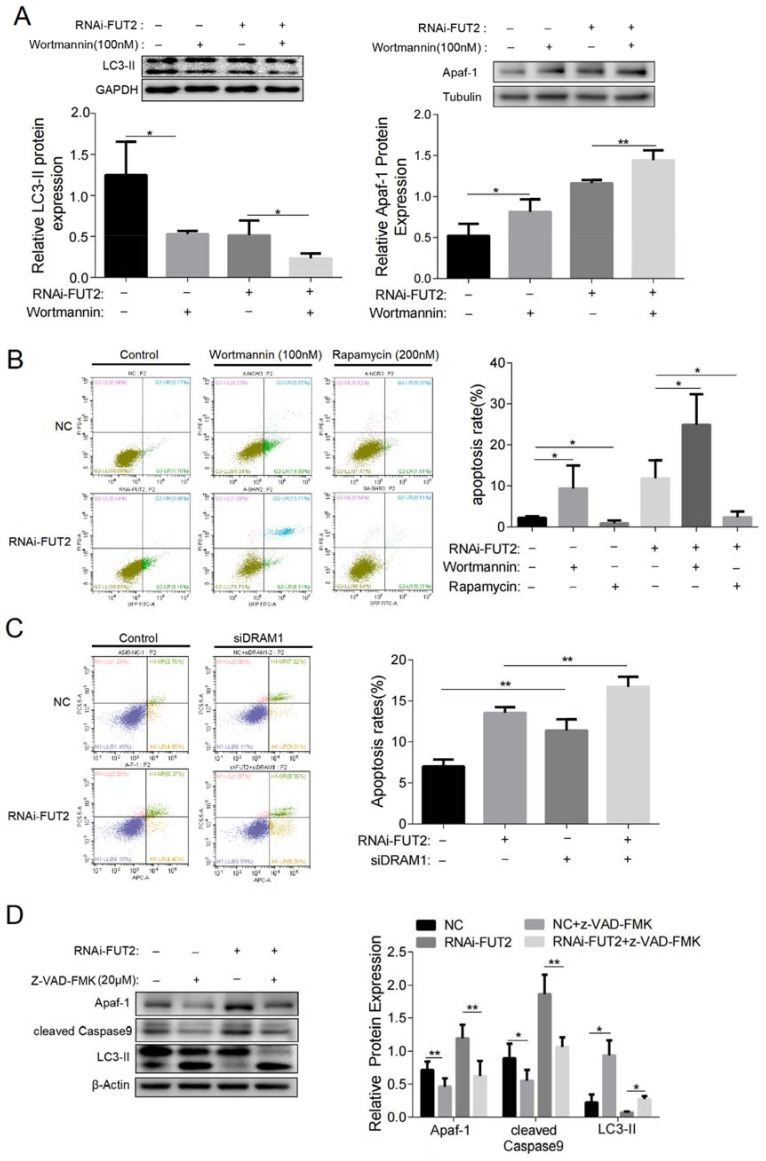
The regulation of autophagy and apoptosis by FUT2 and their antagonism in LUAD cells. (**A**) Representative Western blot showing the expressions of LC3-II and Apaf-1 in FUT2-knockdown A549 cells or control cells in the presence or absence of wortmannin. (**B**) Flow cytometry was used to analyze the apoptosis rate in FUT2-knockdown A549 cells or control cells in the presence or absence of wortmannin or rapamycin. (**C**) Flow cytometry was used to analyze the apoptosis rate in FUT2-knockdown A549 cells or control cells in the presence or absence of siDRAM1. (**D**) Representative Western blot showing the expressions of Apaf-1, cleaved caspase 9, and LC3-II in FUT2-knockdown A549 cells or control cells in the presence or absence of z-VAD-FMK. (* *p* < 0.05, and ** *p* < 0.01).

**Figure 8 cells-11-04031-f008:**
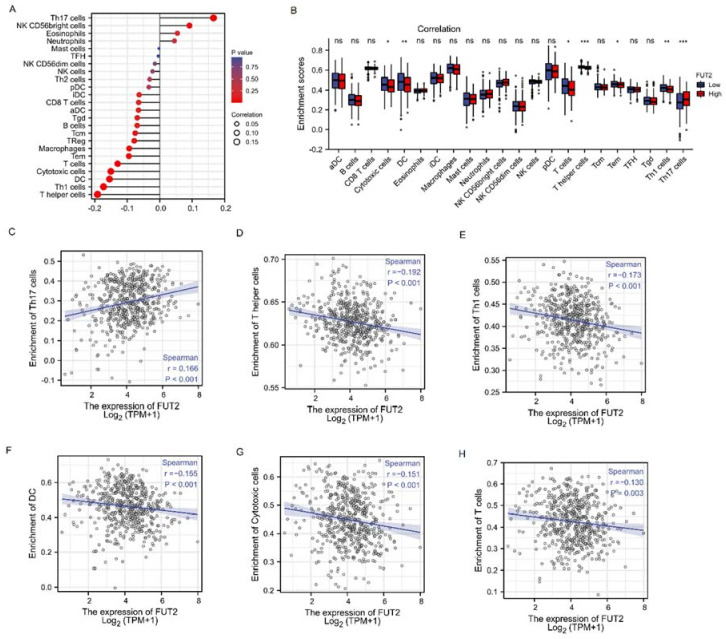
The association between FUT2 expression and immune infiltration in lung adenocarcinoma. (**A**) Correlation analysis between FUT2 and the 24 immune cell types. (**B**) Correlation analysis the different expression levels of FUT2 in 22 immunocyte types. (**C**–**H**) Pearson correlation analyses between FUT2 expression and the enrichment of immune cells, including Th17, T helper, Th1, DC, cytotoxic, and T cells. (ns: No significance, * *p* < 0.05, ** *p* < 0.01, and *** *p* < 0.001).

**Table 1 cells-11-04031-t001:** Primer sequences used for gene expression analysis.

Gene	Forward Primer	Reverse Primer
FUT2	5′-GTGGTGTTTGCTGGCGATGG-3′	5′-AAAGATTTTGAGGAAAGGGGAGTCG-3′
Beclin1	5′-AACCAGATGCGTTATGCCC-3′	5′-ATTGATTGTGCCAAACTGTCC-3′
LC3	5′-AACATGAGCGAGTTGGTCAAG-3′	5′-GCTCGTAGATGTCCGCGAT-3′
P62	5′-AGCGTCAGGAAGGTGCCATT-3′	5′-CCTTTCTCAAGCCCCATGTTG-3′
GAPDH	5′-GAACATCATCCCTGCCTCTACT-3′	5′-CCTGCTTCACCACCTTCTTG-3′

**Table 2 cells-11-04031-t002:** The relationship between FUT2 mRNA expression and clinical parameters of patients with lung adenocarcinoma.

Characteristic	Low Expression of FUT2	High Expression of FUT2	*p*
*n*	267	268	
T stage, *n* (%)			0.001
T1	101 (19%)	74 (13.9%)	
T2	143 (26.9%)	146 (27.4%)	
T3	13 (2.4%)	36 (6.8%)	
T4	8 (1.5%)	11 (2.1%)	
N stage, *n* (%)			0.522
N0	178 (34.3%)	170 (32.8%)	
N1	44 (8.5%)	51 (9.8%)	
N2	36 (6.9%)	38 (7.3%)	
N3	0 (0%)	2 (0.4%)	
M stage, *n* (%)			0.936
M0	177 (45.9%)	184 (47.7%)	
M1	13 (3.4%)	12 (3.1%)	
Pathologic stage, *n* (%)			0.159
Stage I	156 (29.6%)	138 (26.2%)	
Stage II	51 (9.7%)	72 (13.7%)	
Stage III	39 (7.4%)	45 (8.5%)	
Stage IV	14 (2.7%)	12 (2.3%)	
Residual tumor, *n* (%)			0.839
R0	179 (48.1%)	176 (47.3%)	
R1	5 (1.3%)	8 (2.2%)	
R2	2 (0.5%)	2 (0.5%)	

## Data Availability

The datasets used and/or analyzed during the current study are available from the corresponding author upon reasonable request.

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
