# Peer review of "FUT2 Facilitates Autophagy and Suppresses Apoptosis via p53 and JNK Signaling in Lung Adenocarcinoma Cells"

_cells, 2022, doi:10.3390/cells11244031_

Round 1

Reviewer 1 Report

Fucosyltransferase 2 (FUT2) in lung adenocarcinoma has been investigated by Zhou et al. (Oncotarget. 2017 Nov 14; 8(57): 97246–97259.). They find growth inhibition and apoptosis induction after FUT2 knockdown. In this study, the authors aimed to investigate the interaction of autophagy and apoptosis induced by FUT2 inhibition. There are several comments to the authors.

1.    A graphical abstract is suggested.

2.    In the abstract, the author concluded that FUT2 might be a new diagnostic marker and therapeutic target of lung adenocarcinoma. However, FUT2 is not a “new” marker since it has been published by Zhou.

3.    Most studies were performed only in A549 cells. The authors should provide experiments from another cell line to convince the readers.

4.    The overexpress study should be performed in H460 cells which showed a less expression of FUT2.

5.    Total PI3KIII should also be shown in Figure 4A.

6.    The unit of scale bar in Figure 5D was wrong.

7.    The authors claimed that FUT2 can inhibit the phosphorylation of JNK leading to suppress the apoptosis of A549 cells. SP600125 should be applied to evaluate the apoptosis by flow cytometry while FUT2 was inhbition.

8.    The blot of LC3 in Figure 7D should be improved. The baseline level of LC3-II was too weak since the authors claimed it was high in A549 cells.

9.    What is the role of the interaction between FUT2 and p53 or JNK should be discussed.

Author Response

Dear  Reviewer:

 Thank you for your  comments concerning our manuscript.Those comments are valuable and very helpful for revising and improving our paper, as well as the important guiding significant to our research. We have studied comments carefully and have made correction which we hope meet with approval. The  responds to your comments are as flowing:

 Fucosyltransferase 2 (FUT2) in lung adenocarcinoma has been investigated by Zhou et al. (Oncotarget. 2017 Nov 14; 8(57): 97246–97259.). They find growth inhibition and apoptosis induction after FUT2 knockdown. In this study, the authors aimed to investigate the interaction of autophagy and apoptosis induced by FUT2 inhibition. There are several comments to the authors.

  1. A graphical abstract is suggested.

Response: Thanks for your suggestion. A graphical abstract is added.

  1. In the abstract, the author concluded that FUT2 might be a new diagnostic marker and therapeutic target of lung adenocarcinoma. However, FUT2 is not a “new” marker since it has been published by Zhou.

Response: Thanks for your comment. It has been revised in the manuscript.

  1. Most studies were performed only in A549 cells. The authors should provide experiments from another cell line to convince the readers.

Response: In our previous studies, A549 and H1299 cells were used to investigate the roles of FUT2 in lung adenocarcinoma cells. In the present study, we focus on the underlying mechanism of FUT2 in lung adenocarcinoma cell, so we only selected the A549 cell line as a model.

Thanks for your suggestion, and it is very helpful for our further study.

  1. The overexpress study should be performed in H460 cells which showed a less expression of FUT2.

Response: Thanks for your suggestion. In the present study, we used the A549 cell line as a model and performed the overexpression of FUT2 in A549 cells, and the opposite of results were observed, compared with the effects of FUT2 knockdown in A549 cells. 

Your suggestion is helpful for our further study.

  1. Total PI3KIII should also be shown in Figure 4A.

Response: Thanks for your comment. In the present study, we found that knockdown of FUT2 has no effect on the expression of ULK1, however, the level of p-ULK1 was reduced. As we known, PI3KIII is phosphorylated by ULK1, so we detected the level of p- PI3K III, not the total PI3KIII.

  1. The unit of scale bar in Figure 5D was wrong.

Response: Thanks for your comment. We revised the figure.

  1. The authors claimed that FUT2 can inhibit the phosphorylation of JNK leading to suppress the apoptosis of A549 cells. SP600125 should be applied to evaluate the apoptosis by flow cytometry while FUT2 was inhbition.

Response: Thanks for your comment. In this study, we used SP600125, as an inhibitor of the JNK signaling, to investigate whether JNK signaling is involved in the autophagy regulated by FUT2. So, we didn’t detect the apoptosis by the flow cytometry.

  1. The blot of LC3 in Figure 7D should be improved. The baseline level of LC3-II was too weak since the authors claimed it was high in A549 cells.

Response: Thanks for your comment. The figure has been revised.

  1. What is the role of the interaction between FUT2 and p53 or JNK should be discussed.

 Response: Thanks for your comment. We have revised that in the section of results.

 Thank you for your precious comments and suggestions. Those comments are all valuable and very helpful for our further study.

Reviewer 2 Report

The manuscript entitled “FUT2 Facilitates Autophagy and Suppresses Apoptosis via p53 and JNK Signaling in Lung Adenocarcinoma Cells” submitted by Zhang et al. is very interesting, however, there are some aspects that should be clarified:

1.       The authors performed gene expression analysis (RTqPCR) using GAPDH as internal reference. International guidelines on quality recommend using at least two to three internal references to avoid bias. Could the authors add more internal references?

2.       The authors did not state how many replicates they used for RTqPCR? Please add this information.

3.       The authos compared FUT2 expression with T-, N-, M- and R-status and stage. How about patient outcome: Is there a correlation of FUT2 expression and survival?

Author Response

Dear reviewer,

Thank you for your comments concerning our manuscript. Those comments are valuable and very helpful for revising and improving our pape. We have studied comments carefully and have made correction which we hope meet with approval. The main corrections in the paper and the responds to the reviewers’ comments are as flowing:

The manuscript entitled “FUT2 Facilitates Autophagy and Suppresses Apoptosis via p53 and JNK Signaling in Lung Adenocarcinoma Cells” submitted by Zhang et al. is very interesting, however, there are some aspects that should be clarified:

  1. The authors performed gene expression analysis (RTqPCR) using GAPDH as internal reference. International guidelines on quality recommend using at least two to three internal references to avoid bias. Could the authors add more internal references?

Response: Thanks for your comment. Except GAPDH, b-actin was also used as the internal reference. However, for each independent experiment, we only used one internal reference. 

  1. The authors did not state how many replicates they used for RTqPCR? Please add this information.

Response: Thanks for your comment. The polymerase chain reaction (PCR) was performed in triplicate for each sample, and which all experiments were repeated three times. We have revised that in manuscript.

  1. The authos compared FUT2 expression with T-, N-, M- and R-status and stage. How about patient outcome: Is there a correlation of FUT2 expression and survival?

Response: Thanks for your comment. In our previous study (Zhou et al., Oncotarget 2017, 8, 97246-97259), we have showed that the elevated expression of FUT2 mRNA was significantly associated with poor overall survival in all lung adenocarcinoma cancer patients, and the expression of FUT2 was correlated with progression-free survival within all lung adenocarcinoma cancer patients. So, we didn’t discuss that in this manuscript.

Thank you for the time you took to review our manuscript and for the comments, which are helpful for our further study.

Reviewer 3 Report

I find the article very interesting for the reader, no major corrections or comments.  English spell check may be required.

I am not against its acceptance, in his current state.

Author Response

Dear reviewer,

Thank you for the time you took to review our manuscript.

Round 2

Reviewer 1 Report

Although the authors revised the manuscript, the major issue is still unsolved. Data from one cell line is missing reliability.

1.     Most studies were performed only in A549 cells. The authors should provide experiments from another cell line to convince the readers.

2.     The overexpress study should be performed in H460 cells which showed a less expression of FUT2.

3.     In the result section 3.9., the authors claimed that FUT2 can inhibit the phosphorylation of JNK leading to suppress the apoptosis of A549 cells.  SP600125 should be applied to evaluate the apoptosis by flow cytometry while FUT2 was inhibition.

Author Response

Dear Reviewer:

  Thank you for your suggestions concerning our manuscript entitled “FUT2 Facilitates Autophagy and Suppresses Apoptosis via p53 and JNK Signaling in Lung Adenocarcinoma Cells”. Those suggestions are valuable and very helpful for revising and improving our paper, as well as the important guiding significant to our research. The main corrections in the paper were marked as red. The responds are as flowing:

Although the authors revised the manuscript, the major issue is still unsolved. Data from one cell line is missing reliability.

  1. Most studies were performed only in A549 cells. The authors should provide experiments from another cell line to convince the readers.

Response: Thanks for your suggestion. We have run the experiments in H1299 cells, and the related data have been added in the Figure 2 and Figure 6. The manuscript was revised.

  1. The overexpress study should be performed in H460 cells which showed a less expression of FUT2.

Response: Thanks for your suggestion. The overexpress of FUT2 was performed in H460 cells, and the related data have been added in the Figure 2 and Figure 6. The manuscript was revised.

  1. In the result section 3.9., the authors claimed that FUT2 can inhibit the phosphorylation of JNK leading to suppress the apoptosis of A549 cells.  SP600125 should be applied to evaluate the apoptosis by flow cytometry while FUT2 was inhibition.

Response: Thanks for your suggestion. We have ordered SP600125 since we received your suggestion. However, they haven’t yet to deliver the product, and we are not sure when we will receive the product. We are sorry that we are not able to provide the results by now, and the extended deadline for submission has arrived.

Reviewer 2 Report

The manuscript was revised accordingly.

Author Response

The manuscript was revised accordingly.

Response: Thank you for the time you took to review our manuscript.

Round 3

Reviewer 1 Report

No further questions.